# Oligodendrocytes, the Forgotten Target of Gene Therapy

**DOI:** 10.3390/cells13231973

**Published:** 2024-11-28

**Authors:** Yasemin Ozgür-Gunes, Catherine Le Stunff, Pierre Bougnères

**Affiliations:** 1Horae Gene Therapy Center, University of Massachusetts Chan Medical School, Worcester, MA 01605, USA; yasemin.ozgurgunes@umassmed.edu; 2MIRCen Institute, Laboratoire des Maladies Neurodégénératives, Commissariat à l’Energie Atomique, 92260 Fontenay-aux-Roses, France; catherine.le-stunff@inserm.fr; 3NEURATRIS at MIRCen, 92260 Fontenay-aux-Roses, France; 4UMR1195 Inserm and University Paris Saclay, 94270 Le Kremlin-Bicêtre, France; 5Therapy Design Consulting, 94300 Vincennes, France

**Keywords:** AAV, gene therapy, oligodendrocytes, central nervous system

## Abstract

If the billions of oligodendrocytes (OLs) populating the central nervous system (CNS) of patients could express their feelings, they would undoubtedly tell gene therapists about their frustration with the other neural cell populations, neurons, microglia, or astrocytes, which have been the favorite targets of gene transfer experiments. This review questions why OLs have been left out of most gene therapy attempts. The first explanation is that the pathogenic role of OLs is still discussed in most CNS diseases. Another reason is that the so-called ubiquitous CAG, CBA, CBh, or CMV promoters—widely used in gene therapy studies—are unable or poorly able to activate the transcription of episomal transgene copies brought by adeno-associated virus (AAV) vectors in OLs. Accordingly, transgene expression in OLs has either not been found or not been evaluated in most gene therapy studies in rodents or non-human primates. The aims of the current review are to give OLs their rightful place among the neural cells that future gene therapy could target and to encourage researchers to test the effect of OL transduction in various CNS diseases.

## 1. Introduction

Gene therapy encompasses four basic approaches. Three of these apply to diseases caused by a single gene: (i) replacing a defective gene by a functional one; (ii) silencing a mutated gene whose gain of function is toxic to cells; and (iii) correcting an abnormal gene sequence in diseased cells. The fourth approach, gene addition, aims to transfer an opportunistic gene not physiologically expressed in target cells to counteract a pathogenic mechanism. Gene transfer currently holds out major hope for the future treatment of certain debilitating diseases of the CNS [1,2,3,4,5,6,7,8,9,10,11]. This is the case for common diseases, such as Parkinson’s disease (PD) or Alzheimer’s disease (AD), as well as less frequent or rare diseases, sometimes involving only a single case [12].

While neurons and astrocytes are most often targeted, several reasons may explain why OLs have been left out of most attempts in CNS gene therapy. First, the role of OLs in many diseases is just beginning to emerge, as discussed in Section 3. Another explanation is that most gene therapy studies have used adeno-associated virus (AAV) vectors equipped with CAG, CBA, CBh, and CMV promoters. Although considered ubiquitous, these promoters do not activate transgene expression within OLs for unknown biological reasons, unless high doses of a vector are administered [13]. Indeed, in almost all gene therapy studies in rodents or non-human primates (NHPs) using such promoters, OL transduction was either absent [14,15,16], not mentioned [15,17,18], or apparently discordant, as discussed in Section 5.1. Based on these observations, OLs have sometimes been wrongly considered resistant to AAV gene therapy.

This review focuses on AAV vectors, which are at the forefront of advances in gene therapy in CNS diseases. More generally, AAV vectors have become the leading gene therapy platform, the basis of hundreds of clinical trials, and a multi-billion dollar industry [19,20,21,22,23,24,25,26], leading to a robust therapeutic pipeline [27,28]. This review will not discuss the enormous potential of gene-editing technology [29], as its application to CNS diseases is only emerging.

## 2. A Brief Overview of Oligodendrocyte Functions

The 17.4 million OLs represent 20% of the cells populating the adult mouse brain [30]. In the human brain, the total number of neocortical OLs is 21–29 billion [31], with nearly 5 billion in the corpus callosum [32]. The number of OLs increases linearly from 7 to 28 billion during the first three years of life, totaling two-thirds of the adult OL population [33]. Although heterogeneous [34,35,36,37,38,39,40] and becoming regionally diverse with age [38], the OL population is remarkably stable in humans, with an annual exchange of only 1/300 OLs, constant from 5 years of age, a 100-fold lower renewal rate compared to mice [32]. This extremely low OL turnover rate is favorable to AAV gene therapy because it allows transgene copies to persist durably in these rarely dividing cells. A major role of OLs is the synthesis of myelin and enwrapment of axons with myelin sheaths. OL turnover contributes minimally to myelin modulation in human white matter, which is mainly carried out by mature OLs [32]. A single OL myelinates 20–60 axons [41] and the energetic demand for this process of myelination in early life is enormous. OLs meet this internal demand through glucose and fatty acid oxidation [42] performed in numerous mitochondria [43] that diminish once myelination is completed. Myelin sheaths continue to turn over in adult life to maintain the balance between myelin synthesis and degradation [44]. Activity-regulated myelination appears to play a major role in tuning the function of neural networks and maladaptive myelination could contribute to disease mechanisms by promoting pathological patterns of neuronal activity [45].

The other prominent role of OLs is the supply of energy substrates—lactate, glucose, ketones—to axons through myelinic channels [46] to feed oxidative axon metabolism [47,48,49] that neurons cannot ensure. Throughout the whole lifespan, long axons require an abundant and steady supply of metabolites from external sources to maintain signal transduction. Both large axonal tracts that run in the optic nerve or motor or sensory tracts of the spinal cord and small caliber axons are vulnerable to energy deprivation [47]. OLs and axons also have a reciprocal signaling relationship in which OLs receive cues from axons that direct their myelination, and OLs subsequently shape axonal structure and conduction. OLs detect fast axonal spiking through K+ signaling, making acute metabolic coupling possible and adjusting the axon–OL metabolic unit to meet axonal demand [50]. In turn, metabolically active OLs contribute to information processing in addition to speeding up conduction velocity [51]. Neuronal activity causes the axonal release of glutamate, inducing NMDA receptor signaling in OLs, which leads to surface expression of the glucose transporter GLUT1 and an increased glycolytic flux that increases the supply of pyruvate and lactate to axons via monocarboxylate transporters (MCT) [47]. Glutamate also stimulates glucose uptake from OLs to axons [52], the release of OL exosomes and the transfer of SIRT2 to the axonal compartment, where deacetylase stimulates mitochondrial ATP generation [47]. A high content of peroxisomes provides OLs with an energy source from fatty acid β-oxidation [42] and with essential neuroprotection against axon degeneration and neuroinflammation [53].

Besides their close structural and functional interactions with axons, OLs have numerous interactions with other glial cells. Indeed, O initiate crosstalking with astrocytes via direct cell–cell contact as well as via secreted cytokines, chemokines, exosomes, and signaling molecules [54,55]. OLs also crosstalk with microglia, have immune functions, express a wide variety of innate immune receptors, and produce and respond to chemokines and cytokines that modulate immune responses of the CNS [56]. Indeed, OLs interact with neighboring immune cells by expressing cytokines (IL-1b, IL-17A, CCL2), chemokines (C-X-C motif chemokine ligands, CXCL-1, CXCL10, CXCL12), antigen-presenting molecules (MHC-I, II), complement and complement receptor molecules (C1, C1q, C3) and complement regulatory molecules (CD46, CD55) [57,58,59,60]. OLs also express the chemokine receptors CXCR1, CXCR2, CXCR3 and CXCR4 [61,62,63]. Through these multiple effectors and pathways, OLs contribute to protecting CNS homeostasis and neuronal integrity from neuroinflammation, demyelination, and degeneration [60,64,65,66]. Crosstalks also exist between OLs and cerebral endothelium [67].

## 3. Contribution of Oligodendrocytes to CNS Diseases

Since CNS diseases are numerous and diverse, this review will illustrate Ols’ contribution with some examples. These examples include defects in single Mendelian genes known to cause pathology through loss-of-function (Canavan disease, X-linked adrenoleukodystrophy, X-ALD, spinal muscular atrophy, SMA) or gain-of-function (Huntington’s disease, HD) that affect cell types where the function of the normal gene product is physiologically important. In another broader category of non-Mendelian CNS diseases, the dysfunction or loss of distinct neuronal populations is due to a combination of multigenic predisposition and yet unknown environmental factors. This is the case for sporadic PD, multiple system atrophy (MSA), AD, amyotrophic lateral sclerosis (ALS), and multiple sclerosis (MS), while psychiatric diseases are grouped in a distinct category of CNS disorders for which the molecular/cellular substratum remains unknown. This review will also briefly cite hypoxic–ischemic disorders occurring in premature newborns and old adults, as well as traumatic spinal cord injury.

We still do not know in which cell type(s) the primary mechanisms that cause a specific CNS disease are initiated. For most diseases, this question remains unanswered, even when the causal defect is known, as in monogenic disorders. The date of onset of the deleterious processes at the cellular level is also unknown. As astonishing as it seems, although PD, AS, and HD are thought of as late-onset diseases, they might instead be considered neurodevelopmental disorders with an early embryonic onset [68].

Given the complexity of the brain microenvironment, it is not easy to implicate OLs in the pathophysiology of most neurodegenerative diseases (NDDs). It is even more difficult, however, to exonerate them, given their close interactions with axons and other glial cells across the various stages of CNS development and aging. Indeed, OL dysfunctions can alter axon myelination in early life, deprive neuro-axonal metabolism of precious energy substrates in the mature brain, or disturb the multiple crosstalks of OLs with other glial cells over the whole life.

The current review focuses on some CNS disorders involving or possibly involving OLs in the development of their shared [69] or specific pathology. Stricto sensu, a primary oligodendrogliopathy, describes a disease caused by a dysfunction of OLs.

*Canavan disease (CD)*, due to homozygous mutations of aspartoacylase (*ASPA*), is to our knowledge the only primary oligodendrogliopathy where the primum movens of the pathology is undisputably located in OLs. Indeed, N-acetylaspartate (NAA), one of the most abundant molecules in the CNS, is synthesized in neurons and hydrolyzed by ASPA in normal OLs [70]. ASPA deficiency thus leads to a direct OL dysfunction inducing NAA accumulation, lack of myelin, and spongiform changes in deep brain structures and cerebellum.

OLs are suspected to contribute to or to aggravate disease mechanisms for many NDDs [71,72,73,74,75], which can also be considered as oligodendrogliopathies.

*MSA* is the most obvious example of such diseases [76,77] and the one for which the term oligodendrogliopathy was first used [78]. The pathognomonic feature of MSA is the accumulation of insoluble protein aggregates mainly consisting of α-synuclein (α-syn) in brain OLs [79,80,81], which leads to severe neuronal loss [74,82,83]. It is not known whether this accumulation is primary or results from a transfer from neurons, where aggregates are also observed in smaller numbers [84,85]. Forced overexpression of α-synuclein in OLs using transgenesis (mice) [86,87,88,89,90,91,92,93] or viral transfer (NHP) [94,95] under the control of a MBP, PLP or CNP promoter produces diseases that share features with human MSA [96,97,98]. Inoculation of brain homogenates from MSA patients to animals triggers the accumulation of protein aggregates in neurons and OLs [99,100,101]. Cytokine profiling in brain tissue from patients showed that pro-inflammatory pathways are upregulated [102]. Whether primary or secondary, the accumulation of aggregates in OLs is associated with the localization of lesions and clinical symptomatology of MSA [103,104,105].

Synuclein-rich protein aggregates are also observed in OLs in PD [106,107] and in various sporadic and genetic tauopathies [74]. OL aggregates are also present in AD [74], ALS [74], and fronto-temporal degeneration linked to TDP43 proteinopathy [74]. Aggregates could impair the myelination of axons, the transfer of energy substrates to neurons and axons, and the interactions of OLs with astrocytes.

Other CNS diseases have also been associated with OL disorders.

In *X-ALD*, the cause of pathology is the impairment of very long chain fatty acid (VLCFA) transport into peroxisomes due to mutations in the *ABCD1* gene, which encodes the ALDP. Pathology is thus suspected to affect cell types whose peroxisomes cannot transport VLCFAs, notably the peroxisome-rich OLs. Spinal cord axons are affected, notably the spinocerebellar and corticospinal tracts [108], leading to Adrenomyeloneuropathy (AMN, the adult form of X-ALD). Inflammation and demyelination can occur in other patients, leading to the devastating cerebral form of X-ALD (CCALD). The same *ABCD1* gene defect can lead to AMN or CCALD, a yet unsolved mystery.

*Spinocerebellar ataxia (SCA)* (*SCA3* being the most common) are also diseases of spinocerebellar tracts and/or cerebellar degeneration [109]. SCA are associated with OL signatures [110].

*Hereditary spastic paraplegias* (HSPs) are heterogeneous group of rare axonopathies. More than fifty Mendelian genes have been identified, but the causality remains unknown in over 30% of cases, even more so for sporadic cases [111,112]. HSPs result from genetic alterations causing dysfunction of the long axons in the corticospinal tract and posterior columns of the spinal cord. Given the close interactions of OLs with the long and vulnerable axons of these tracts, transducing spinal cord OLs with metabolic transgenes could contribute to improving axonal health.

In *PD*, only a few studies have investigated changes in OLs [74,113]. Yet such studies would be important since OLs show specific molecular signatures in PD patients, including pathways implicated in inflammation and myelination [114]. Actually, sporadic PD continues to face multiple knowledge gaps and many pathological mechanisms remain to be identified [115].

In *Alzheimer’s disease (AD)*, degeneration of white matter and demyelination appear to be important features, supporting a pathogenic relationship between OL dysfunction and AD hallmarks [74,116,117,118,119,120,121,122,123].

In *Huntington’s disease (HD)*, huntingtin overexpression can affect a variety of cells including OLs [124,125]. Mutant huntingtin is expressed in OLs, reducing myelin gene expression and causing an age-dependent demyelination and symptoms in HD mice [126]. Several studies suggest that dysfunction of mature OLs is involved in HD through early myelin pathology [124,126,127,128,129,130].

In *ALS*, OL abnormalities were reported in patients and rodent models [131,132,133,134], possibly leading to impaired trophic support to axons and weakening the already vulnerable motor neurons [135].

In *MS*, pathological processes including neuroinflammation and energy failure cause dysfunction and apoptosis of OLs, leading to demyelination and neurodegeneration [122,136,137,138,139,140].

The two main phenotypes of *hereditary optic neuropathies,* dominant optic atrophy and Leber optic neuropathy, appear to involve mitochondrial dysfunction in retinal ganglion cells and their axons (newman, biousse carelli). The metabolic support by OLs to the large axonal tracts that run in parallel in the optic nerve is critical [42,47]. Demyelinating optic neuritis is an inflammatory optic neuropathy [141].

In several *psychiatric disorders*, a significant involvement of OLs is suspected [142,143], notably in autism [144,145,146,147,148] and depression [149,150,151]. It is noteworthy in this respect that dysfunction of OLs resulting from *CHD8* haploinsufficiency gives rise to autistic phenotypes in humans and mice [152].

In *perinatal diffuse white matter injury* (periventricular leukomalacia, hypoxic–ischemic encephalopathy), the most common type of brain injury in preterm infants, a multifactorial interplay of events causes OL death, leading to failure of myelination in the developing white matter [153]. Mature OLs are relatively preserved following inflammatory and/or hypoxic–ischemic damage, compared with late-stage OL progenitors (OPCs) [153].

*Brain ischemia–reperfusion injury* affects OLs through oxidative stress, inflammation, mitochondrial dysfunction and excitotoxicity, generating demyelination, axonal function and survival [154,155].

*Vascular dementia* (VaD) involves a variety of neuronal and vascular lesions, leading to oxidative stress, inflammatory damage and demyelination, and it is closely associated with OL dysfunction [156].

*Traumatic spinal cord injury* triggers OL necrosis and apoptosis, with apoptosis continuing at the chronic stages of evolution. Loss of OLs causes demyelination and impairs axon function and survival [157,158].

## 4. General Principles of Gene Therapy for CNS Diseases

The main mechanisms underlying CNS diseases involve metabolic dysfunction, defective myelination, neuroinflammation, neuronal or axonal death. These deleterious mechanisms operate at a local or more general level of the brain or spinal cord and can act alone or combine their effects at different stages of disease evolution. Their endpoint is neuron pathology in different functionally and spatially defined CNS regions, such as the striatum, substantia nigra, or hippocampus, or in a more diffuse process affecting various zones simultaneously or successively.

Gene therapy holds promise for treating severe disorders by delivering a cargo of therapeutic genetic material to specific cell types. As previously pointed out, this review focuses almost exclusively on AAV vectors [159,160]. General information about AAV gene therapy can be found in a number of excellent reviews [1,10,22,29,161,162,163,164,165]. More specifically, AAV vectors have emerged as prominent tools for transferring genes to the CNS [4,9,22,23,24,26,166]. Successful transduction by AAV vectors is contingent on many key steps such as cell surface receptor binding, endocytic uptake, endosomal escape, nuclear entry, capsid uncoating, genome release, second strand synthesis and subsequent transcription and translation [162,167,168,169]. Within the last ten years, the number of trials using rAAV vectors for deliver therapeutic genes to the CNS has grown rapidly.

During the neonatal period, several intravenously delivered (iv) AAV serotypes can cross the BBB and reach neural cells in the CNS of rodents, non-human primates (NHPs), and humans [13,14,15,18]. At later ages, the iv route cannot achieve a significant transgene expression in the adult CNS, unless a high vector dose is administered [170,171,172,173], exposing liver cells to a heavy load of capsids and transgene copies [174,175,176,177,178]. AAV vectors can alternatively be delivered directly to the CNS using intraCSF injections into cerebral ventricles (intracerebroventricular, icv), cisterna magna (icm), or intravertebral lumbar puncture for intrathecal (it) administration. Vectors can also be administered with stereotaxic intraparenchymal injections directly into the hippocampus, thalamus, cortex, or striatum [179]. When injected into the striatum for example, the spread of an AAV vector expression extends over 1–2 mm [180]. The capsid serotype and age of the animal affect the spread of the vector from the injection site into distinct regions of the brain or spinal cord [181].

The timing of vector administration is another factor that can greatly impact the efficacy of gene therapy. For example, preclinical and clinical studies of SMA gene therapy have led to better therapeutic efficacy in younger patients [182]. Unfortunately, the precise translation of the optimal treatment timing from animal models to patients is often challenging, due to differences in disease mechanisms, CNS development, and lifespan across species. A particular and rare case is that of diseases that are identified near birth thanks to neonatal screening, opening the door to a preventive effect of gene therapy for diseases that manifest later in life, such as AMN.

The AAV capsid should be able to reach and transduce targeted neural cells. The serotype can influence its ability to do so. Indeed, capsid composition and structure determine cell surface receptor binding. The AAV9 capsid serotype, which enters CNS neurons, astrocytes, OL, and pericytes, has been widely used to target neural cells in rodent or non-human primate studies [13,14,15,18,183], as well as in clinical trials for SMA and PD. Capsid composition can be manipulated by introducing specific mutations to achieve greater transduction levels, decrease immunogenicity and increase cell specificity [184]. In the past few years, new neurotropic capsids have emerged [21,184,185,186,187], including oligotropic capsids [95,188,189,190], which will be discussed in Section 6.1. Since the development of potent tissue-specific vectors has become an area of critical need, several laboratories have developed novel recombinant capsid variants [1,4,191,192,193,194]. For example, the capsid variant PHP.B targeting LY6A receptor to carry the vector through the BBB into the brain [187,195] turned out to have a tropism specific to a mouse strain and lacking in other mouse strains or NHPs [196]. PHP.B was used as a backbone to derive a new variant, AAV.CAP-B10, which enabled a robust transduction of neurons in marmoset brain after iv administration and an interesting de-targeting effect from the liver. Diverse AAV serotypes and variants with high retrograde and/or anterograde transport properties have been described in the last few years [197].

The affinity and transduction efficiency of a given AAV variant in different cell types may vary among species. This is why any novel capsid should be tested in both NHPs and mice to better predict its translatability to humans. Translation is particularly challenging when a capsid works in NHPs but not in the mouse disease model.

The choice of a promoter able to activate a therapeutic transgene in specific target cells is critical. The ubiquitous promoters, such as CAG, CBA, CBh or CMV promoters widely used in CNS gene therapy, have a robust track record of efficiency. They are active in neurons and astrocytes but poorly active in OLs. In recent years, cell-specific or cell tropic promoters have emerged in gene therapy [19,21] to ensure expression in a given cell type. Such regulatory cassettes need to induce a therapeutic level of transgene expression without having to use a potentially toxic vector dose. Promoters designed to target specific subsets of cells have shown success in mouse or primate studies, but to our knowledge, no clinical trials using such promoters have yet been performed for CNS diseases.

Adverse events resulting from CNS gene therapy are outside the scope of the current article. For a comprehensive discussion of this topic, refer to an excellent review in the literature in reference [4].

## 5. Gene Therapy Targeting CNS Cell Types Other than Oligodendrocytes

In order to delineate a potential place for OLs in the gene therapy of CNS diseases, this paragraph briefly summarizes the current status of gene transfer experiments targeting other neural cells.

### 5.1. Targeted Cells

*Neurons* are obviously the prominent targets of CNS gene therapy. Since the early 2000s, dozens of preclinical studies in rodent models or NHPs [1,2,3,4,5,6,7,8,9,10,26,198,199,200,201,202,203,204,205,206,207,208,209,210,211,212,213,214,215,216,217,218,219,220,221,222,223,224,225,226,227,228,229,230,231,232,233,234,235] and clinical trials [236,237], too numerous to be all cited here, have focused on neuron transduction by AAV vectors using neurotropic capsids, most often AAV9, and ubiquitous promoters.

To a lesser extent than neurons, microglia and astrocytes have also been targeted in gene therapy in a few animal models and clinical trials.

*Microglia* are actively involved in shaping the brain’s inflammatory response to stress [238]; thus, they are attractive targets for some CNS diseases. However, they are difficult to transduce with AAV vectors, notably because of their low affinity for current capsids, poor activation of the ubiquitous promoters, and active renewal and division that dilute episomal transgene copies [239]. Overall, microglial transduction by the current AAV vectors equipped with ubiquitous promoters has been largely unsuccessful in mouse models [14,181,240,241,242,243,244,245,246,247,248,249,250,251,252,253,254,255,256,257,258,259]. New capsids and more cell-specific promoters are likely to improve transduction [21,252,260], but they will not prevent the mitotic dilution of transgenes during active microglia divisions. As an alternate option, lentiviral (LV) vectors that integrate in the host cell nuclear genome have been used to transduce microglia [242]. This can be achieved in vivo with limited success using direct intracerebral LV injections of lentiviral vectors (LVs). Instead, hematopoietic stem cells (HSCs) from a patient can be transduced ex vivo and then grafted into this myelo-ablated patient to migrate to his brain and differentiate into microglia expressing the desired transgene [261]. This autotransplantation, pioneered in gene therapy trials for CCALD, has been shown to correct neuroinflammation and stabilize demyelination for a few years [262,263]. However, the other ABCD1 mutated cells, neurons, astrocytes, Ols and pericytes remained uncorrected, and the midterm evolution was devastating in these early attempts [264]. In addition the use of LV vectors was recently shown to induce leukemia in 7/67 patients [265]. Microglia can also serve as a vehicle for importing lysosomal enzymes into other brain cells [266].

*Astrocytes* make up 17–61% of the cells in the human brain depending on the area [267] and have also been targeted by several gene therapy approaches. These cells contribute to neurotransmitter cycling, metabolic support of neurons and maintenance of the blood–brain barrier (BBB) [238,268,269]. The complex physiology of astrocytes includes regulation of glutamate and ion homeostasis, cholesterol and sphingolipid metabolism, and responses to environmental factors [270]. Astrocytes also contribute to neurotransmitter cycling, metabolic support of neurons and maintenance of the blood–brain barrier (BBB) [238,268,269] but can loose their supportive functions and become reactive and neurotoxic in pathological conditions [269]. Astrocyte dysfunction occurs in numerous neurological diseases such as PD, AD, HD, MS, and neuropsychiatric disorders. Astrocytes can be transduced by AAV vectors [270,271,272,273,274,275,276,277,278,279,280,281,282,283]. AAV9 vectors equipped with ubiquitous promoters co-transduce varying proportions of astrocytes and neurons in mouse models (254–267). A selective transduction of astrocytes can be achieved using astro-specific promoters such as GFAP [284,285]. Diverse AAV serotypes and variants with high retrograde and/or anterograde transport properties have been described in recent years [197].

*Ependymocytes* [17] *pericytes*, or *endothelial cells* [13,286], have rarely been targeted by AAV gene therapy. Newly engineered AAV9 variants, AAV9-X1 and AAV9-X1.1, have shown specificity and efficiency in targeting endothelial cells, with about 95% specificity across various brain regions in mice, but not in marmosets or rhesus macaques [287].

### 5.2. Examples of Gene Therapies Based on Neuron or Astrocyte Targeting in CNS Diseases

This section presents a non-exhaustive list of CNS diseases in which gene therapy has been aimed at transducing neurons and/or astrocytes. This will introduce the later discussion of an OL-targeting gene therapy approach for the same diseases.

#### 5.2.1. *Canavan Disease* (CD)

*Canavan disease* (CD) is a devastating leukodystrophy starting in early life, caused by mutations in the *ASPA* gene. Currently, no effective treatment is available. The lack of ASPA activity in OLs leads to the accumulation of NAA in the brain, disrupting myelin formation and maintenance. Over the past two decades, several laboratories have attempted gene therapy of CD in animal models, focusing on the transduction of neurons and/or astrocytes (Table 1).

First-generation AAV vectors resulted in incomplete therapeutic outcomes in mouse models [288,289]. In 2013, Gao’s lab demonstrated that a single iv injection of a rAAV9 vector expressing hASPA achieved partial but unsustained rescue of the severe phenotypes of CD knockout (KO) mice [293]. In 2017, the same team designed two vectors containing a codon-optimized hASPA cDNA controlled by either a ubiquitous promoter or a GFAP promoter, which were delivered intravenously (iv) to AKO pups at P1. Astrocyte-restricted hASPA expression improved motor function, pathology, and biomarkers [290]. High-dose iv, but not it or icv, vector administration to adult cynomolgus macaques resulted in transduction and ASPA expression in deep-brain structures [170]. In 2013, an AAV2-ASPA vector was injected into six brain sites in 13 patients [294]. In 2023, D. Gessler and G. Gao reported a 4-year follow up of a unique patient who had received simultaneous iv and icv injection of an rAAV9-CB6-ASPA vector [295]. Two clinical AAV-ASPA trials are currently in Phase I/II. In the CANaspire trial (NCT04998396), an AAV9-based codon-optimized ASPA vector showed a positive impact on biomarkers and MRI results in four patients, as presented orally at the American Society of Gene and Cell Therapy 2023 Annual Meeting.

#### 5.2.2. *Multiple System Atrophy*

The objectives of gene therapy for MSA could include diminishing TNFα-mediated neuroinflammation and its secondary consequences including oxidative stress, glutamate-related excitotoxicity and neuronal loss. Stimulation of neuronal and glial proliferation, enhancement of myelination through trophic support and cell replacement therapies could also be pursued by gene therapy. Although studies in MSA mouse models have identified effectors of the pathophysiological cascade and unraveled secondary changes caused by aberrant α-synuclein aggregation [296,297,298], there have been only few AAV gene therapy attempts in MSA mouse models [297,299]. One of these attempts involved the LV vectors delivering the nuclear-related factor 2 (NRF2), the glutamate dehydrogenase 2 (GDH2), and the excitatory amino acid transporter 2 (EAAT2) genes into the striatum of two mouse models, resulting in a significant improvement in motor function [300]. A Phase 1 trial (Regenerate MSA-101) is underway to evaluate the effects of AB-1005 (AAV2-GDNF) in patients affected by MSA-parkinsonian type (MSA-P) (*NCT05167721. GDNF Gene Therapy for Multiple System Atrophy*). This vector delivers glial cell line-derived neurotrophic factor (GDNF) directly to the putamen, aiming to preserve dopamine neurotransmission that is reduced in MSA-P.

#### 5.2.3. *Adrenomyeloneuropathy* (AMN)

AMN is a « pure » axonopathy caused by *ABCD1* mutations that affects the long axons of the descending and ascending tracts of the spinal cord in mid-adulthood [108]. OLs provide myelin sheaths and energetic support to these axons. A pioneering gene therapy attempt utilized an AAV9-CBh-hABCD1 vector in 6-week-old *Abcd1−/−* mice [301]. Fifteen days after intravenously injection, this vector had transduced 23% of neurons, 18% of astrocytes and 7% of OLs, leading to a 12–24% decrease in VLCFA accumulation, but did not show a clear effect on motor function [301,302]. In another experiment, an rAAV9-CBA-hABCD1 vector was injected intrathecally, resulting in transduction of neurons, astrocytes and endothelial cells, but no OLs, in the spinal cord of *Abcd1−/−* mice [303]. Despite the absence of clinical improvements in these mice, a clinical trial was launched using injection of the AAV9-CBA-ABCD1 vector (NCT05394064 sponsored by SwanBio).

#### 5.2.4. *Parkinson’s Disease*

We will briefly summarize the main gene therapy attempts to illustrate the heterogeneity of animal studies in this disease, describe the main transgenes that have been tested, and analyze the translation of experimental results to patients. The delivery of genes coding for GDNF [304,305,306,307], neurturin [308,309,310] or CDNF [311] has shown promise in rodent models of PD. An LV vector carrying GDNF was injected into the striatum and substantia nigra of young adult rhesus monkeys treated with 1-methyl-4-phenyl-1,2,3,6-tetrahydropyridine (MPTP) or of non-lesioned aged monkeys. This gene therapy prevented nigrostriatal degeneration and was said to induce some local regeneration [312]. AAV-based gene therapies in mouse or primate models of permanent dopaminergic depletion targeted nigro-striatal neurons with the aim of improving dopamine synthesis by delivering genes encoding glutamic acid decarboxylase (GAD) [313], aromatic L-amino acid decarboxylase (AADC) [224], tyrosine hydroxylase (TH) and GTP cyclohydrolase (CH1) carried by separate [314] or single AAV vectors [315] equipped with Syn1 or CMV promoters [316]. Two LV vectors, Prosavin [317] and OXB-102 [210,318], were used for improving dopamine synthesis by delivering *AADC*, *TH* and *CH1* genes to MPTP-lesioned cynomolgus macaques. Another gene therapy strategy aimed at transducing striatal D1 medium spiny neurons (MSNs) was able to modulate dysregulated neural circuitry and improve motricity in mouse and primate PD models [319].

Fifteen trials based on three strategies have now been conducted including over 400 patients with PD [320], the highest recruitment in the field of CNS gene therapy. The first strategy is the delivery of genes encoding a neurotrophic factor to nigral dopaminergic neurons [321]. Another strategy involves modulating basal ganglia outputs by delivering glutamic acid decarboxylase (GAD) to the subthalamic nucleus [313,322]. The third category of gene therapy trials for PD is based on the transfer of the AADC gene to enhance conversion of L-Dopa to dopamine by nigro-striatal neurons before the occurrence of neuronal loss [209,323]. More recently, gene therapy trials tested the ProSavin LV vector [324,325] and then the optimized OXB-102 LV vector (under the name AXO-Lenti-PD) [318].

Although all of these Phase I trials initially reported encouraging efficacy results, follow-up studies showed no or modest motor improvements.

#### 5.2.5. *AD*, *HD*, *ALS*, *SMA*

AAV vectors targeting neurons and or astrocytes have been tested in AD [230,326], HD [124] and ALS [327,328,329]. Various transgenes, such as APOE2 [330], nerve growth factor (NGF) [331,332] or cholesterol-24-hydroxylase [333], have been tested in AD.

In SMA, preclinical studies resulted in one of the few approvals of AAV gene therapy for human use, with currently positive clinical effects [177,334,335,336].

### 5.3. A Global Overview of Clinical Trials of CNS Gene Therapy

Eighty-seven clinical trials have already used the administration of rAAV vectors for the treatment of CNS diseases [236,237]. Among the genetic diseases mentioned are Adrenomyeloneuropathy (AMN), CD, SMA (15% of trials), HD, Giant Axonal Neuropathy, IGHMBP2-Related Diseases, Late Infantile Neuronal Ceroid Lipofuscinosis, Menkes Syndrome and spastic paraplegia type 50. These disorders are caused by single gene mutations. Another group of more common non-Mendelian diseases comprises PD (18% of trials), MSA, AD, Frontotemporal Dementia (FTD) and ALS. Other diseases include storage disorders and various neurological conditions such as Rett Syndrome and temporal lobe epilepsy. The main ROAs were intraparenchymal (nearly 50%), notably striatal (38%), intravenous (23%) and intrathecal (27%). Soon after preclinical studies [337], a 4-year-old single patient with hereditary spastic paraplegia type 50 (SPG50) received an AAV9-AP4M1 vector intrathecally [12].

It is remarkable that only 1 out of 87 of the cited clinical trials targeted OLs: this unique trial is NCT04833907 (sponsored by Myrtelle), which uses icv administration of an Olig001-CBh-ASPA vector.

## 6. OL as Attractive Targets for CNS Gene Therapy

As long as ubiquitous promoters were used to equip AAV9 vectors, neurons and astrocytes were abundantly transduced, while the capacity of vectors to transduce OLs remained in question. For example, high intravenous doses of an AAV9 vector equipped with a CMV promoter were able to transduce many OLs in the brain and spinal cord of adult mice and marmoset, together with neurons and astrocytes in adult mice [13]. (Note that transduction was evaluated only two weeks after vector injection). In contrast, very high doses of a comparable vector iv injected to mouse neonates at P1 did not transduce OLs significantly, presumably because OL progenitor cells (OPCs) divide too rapidly at this age to allow persistence of extra-chromosomal gene copies in the oligodendroglial lineage [16,338,339]. In cynomolgus macaques, intravenous rAAV9 transduced brain astrocytes but no OLs [14,15]. OL transduction was not studied after intracerebral injection to adult mice [24], intracisternal injection to cynomolgus macaques [15] or intraputaminal injection to two rhesus macaques [18]. Based on these studies, OLs have sometimes been wrongly considered resistant to AAV gene therapy and were almost completely left behind in the gene therapy pipeline, for reasons that are no longer justified. Indeed, mature OLs hold promise not only for the treatment of CNS diseases considered to involve white matter like CD, X-ALD and MS but also for conditions where pathology of OLs and myelin has been described, such as PD, AD, ALS, HD and psychiatric diseases. It should be noted that mature OLs arise only at P7-P10 in mice, opening the door to durable transgene expression. After this age, the slow turnover of mature OLs allows AAV-carried transgene copies to persist in mouse OLs [340], and presumably in humans whose OL turnover is even slower. For NDDs associated with aging in humans and mice, the durability of transgene expression should be assessed in the long term, while preclinical studies often report transduced cell counts only 2–3 weeks after vector administration [301,303,341], with few exceptions [340].

### 6.1. Oligotropic Capsids and Promoters

Oligotropic vectors are those that can transduce a “fair” percentage of OLs, possibly in association with neurons, astrocytes or microglia, while oligospecific vectors are those that induce transgene expression exclusively (and abundantly) in OLs (not yet reported). Such properties are due to vector capsids and promoters. AAV capsids attach to extracellular glycans (e.g., glycoproteins, glycolipids) prior to cell entry [342,343]. Although OLs synthesize glycans actively for myelin synthesis [344], we have not found information about the glycan composition of OL membranes in a healthy or diseased brain. We were also unable to find information on the AAV receptors present on the OL membrane. All natural AAV capsids do not show primary OL tropism. A small number of natural AAV serotypes can transduce OLs when equipped with ubiquitous promoters, but the transduction efficiency is low [345], as for AAV8 or AAV9 carrying a CMV promoter [346,347] or a CBA promoter [348]. AAVrh10 transduce OLs when equipped with a cytomegalovirus/β-actin hybrid promoter [349]. The AAV-PHP.B with a CAG promoter allows for widespread transduction of the mouse CNS including OLs [187]. Capsid engineering has produced a chimeric capsid capable of transducing both neurons and OLs [348]. Olig001, a variant rAAV capsid made of a chimeric mixture of AAVs 1, 2, 6, 8 and 9, has over 95% specificity for OLs [189], following striatal injection into rats when equipped with a CBA promoter. Following intraparenchymal injection of Olig001-CBh to neonatal and adult nur 7 mice, 90% of transduced cells were OLs [188]. In other experiments, 60–95% of cells transduced by Olig001-CBh were OLs in the spinal cord, striatum and cortex of healthy rodents [95,189] and NHPs [95], except in the cerebellum, thalamus or midbrain, where transgene expression was modest [190].

Few cell-specific promoters have been used for expressing transgenes in OLs [271]. Initial studies used the myelin basic protein (MBP) promoter. A 1.9 kb Mbp promoter was able to drive GFP expression in OLs in the mouse brain but not in neurons, astrocytes or microglia [345,350,351,352]. The stage of development impacts Mbp-driven transgene expression in OLs. While in P10 and P90 mice, the majority of GFP expression was observed in OL, the number of transduced OL was only 3% in P0 mice. A similar pattern of OL transduction was seen when the vector was injected into the brains of P10 mice pups [180]. Different regions of the *Mbp* promoter have shown various levels of OL expression in the postnatal and adult CNS [353]. In the striatum of adult mice, the Mbp promoter exhibited an OL specificity of 91% for rh39 capsids, 87% for cy5 and 78% for AAV 1/2 (78%) [354]. An AAV vector containing the *Mbp* promoter delivered to the internal capsule of P10 Cx32/Cx47 double-knockout mouse improved myelination and reduced OL apoptosis, inflammation and astrogliosis [355]. However, while the Mbp promoter targets OLs, its relatively large size limits its use. The nucleotide sequences of the human and murine MBP proximal promoter regions are approximately 90% identical [356], thus suggesting that the proximal human MBP promoter would be active in rodent cells. To our knowledge, no studies have tested small fragments of the Mbp promoter, particularly those defined in [353] or [357], or enhancer regulatory sequences known to modulate the transcription of the Mbp gene [358]. The MBP promoter has also been used to transduce OLs in NHPs [93,359].

Modification of constitutive promoters can shift gene expression from neurons to OLs [352]. As expected, when an AAV9 vector with a full-length CBA promoter was infused into rat striatum, 88% of transduced cells were neurons. But unexpectedly, a truncated CBA promoter, CBh, induced transgene expression in 38% of OLs and 46% of neurons. Modification of the VP2 region (six-glutamate residue insertion) increased the OL specificity of the full-length CBA promoter up to 80%.

Another promoter, the promoter of the human myelin-associated glycoprotein (*MAG*), exhibited a distinctive capacity to transduce OL. This was the case when using a cy5 AAV vector injected into the striatum of adult mice [354]. Three constructs placed GFP under the control of either a 2.2 kb MAG promoter or truncated 1.5 and 0.3 kb fragments. Transgene GFP expression in OLs represented 98.4% of transduced cells with the 2.2 kb promoter and 90.7% with the 0.3 kb promoter, so that 65% of OLs were transduced with the 2.2 kb promoter and 57% with the 0.3 kb promoter. Interested by its small size, we used the 0.3 kb promoter in *Abcd1−/−* mice, a model of AMN. The AAV9-MAG0.3-ABCD1 vector was injected iv at P10. Among transduced cells of the cervical spinal cord, 68% were OLs, so that 50–54% of all OLs expressed hALDP. In cerebellum, 45% of OLs expressed the transgene, with none in the cerebral cortex. Unexpectedly, 30% of astrocytes also expressed ALDP, while neurons or microglia showed no expression [340]. A modification of the MAG promoter sequence demonstrated increased oligotropic expression in vitro in human oligodendroglioma (HOG) cells in culture when compared with the 0.3 kb promoter sequence (Figure 1).

Oligotropic promoters may also replace the promoters currently used in LV vectors to express therapeutic genes in neurons or astrocytes [360,361].

### 6.2. Diseases That Could Benefit from Gene Therapy Targeting OLs

Several pathological conditions illustrate the potential of OLs as gene therapy targets.

#### 6.2.1. Genetic Leukodystrophies

*Canavan disease (CD*). Given that ASPA expression is restricted to white and gray matter OLs, and OLs are the primary cells affected in CD [362,363], the most logical therapeutic strategy would be to restore ASPA activity in OLs as early in life as possible, using vectors composed of an oligotropic capsid and a promoter capable of expressing the *ASPA* transgene in OLs. An OL-targeting approach has been implemented in two mouse models of CD (Table 2).

In a third study, the Olig001-CBh-*ASPA* vector was icv injected at P2 into *nur7* mice. This resulted in 80–90% oligotropism and 6–20% neurotropism in the cerebral cortex, striatum and spinal cord. Olig001-CBh-*ASPA* transduced approximatively 25% of OLs in the cortex, 10% in the striatum and 31% in the spinal cord [190]. It would be interesting to know the percentage of OLs transduced by this vector in a young baby primate. NCT04833907 (sponsored by Myrtelle) utilized icv administration of an Olig001-CBh-ASPA vector to target OLs in CD patients [295]. To our knowledge, this is the only clinical trial targeting OLs. Ongoing and future studies will explore novel vector designs to enhance OL-specific targeting in CD models. A promising approach under development in our team involves the use of an AAV9 vector equipped with MAG promoter or MAG promoter variants (Figure 1).

*Several other leukodystrophies* could benefit from OL targeting [365] such as Pelizaeus-Merzbacher like disease (PMLD) [355].

#### 6.2.2. *Axonopathies of Spinal Tracts*

*Adrenomyeloneuropathy* (AMN). We have seen above that pioneering gene therapy utilized an AAV9-CBh-hABCD1 vector in *Abcd1−/−* mice [301], leading to the NCT05394064 clinical trial. Since OLs provide myelin sheaths and energetic support to the long axons of the descending and ascending tracts of the spinal cord, we developed an alternate strategy by targeting OLs with the MAG promoter in an AAV9 capsid. Three weeks after iv injection to *Abcd1−/−* pups, this vector induced *ABCD1* expression in 50–54% of OLs and 29–32% of astrocytes. This level of expression in astrocytes was unexpected [354]. Our vector prevented AMN motor deficits for up to 24 months of age. Now that AMN is screened neonatally in a growing number of countries, our results pave the way for a possible prevention of AMN. Since there is no X-ALD mouse model developing brain demyelination, we could not estimate whether the transduction of *ABCD1* in brain OLs would possibly help to prevent CCALD.

*Spinocerebellar ataxia* (SCA) (SCA3 being the most common) are also diseases of spinocerebellar tracts and/or cerebellar degeneration [109]. SCA are associated with OL signatures [110]. Restoring myelination [366] or brain cholesterol turnover are two potential objectives of gene therapy [366,367].

*Hereditary spastic paraplegias* (HSPs) are a heterogeneous group of rare axonopathies.

More than fifty Mendelian genes have been identified, but causality remains unknown in over 30% of cases, even more so for sporadic cases [111,112]. HSP results no! it should be “HSPs result” from genetic alterations causing dysfunction of the long axons in the corticospinal tract and posterior columns of the spinal cord. Given the close interactions of OLs with the long and vulnerable axons of these tracts, transducing spinal cord OLs with metabolic transgenes could contribute to improving axonal health.

#### 6.2.3. *HD and ALS*

*Huntington’s disease* (HD). Targeting OLs is inspired by the demyelination hypothesis of HD, demonstrating aberrant myelination and changes in OLs in HD brain [368,369]. Other gene therapy approaches targeting OLs in HD could address thiamine [125], cholesterol [370] or cysteine metabolism [371].

*Amyotrophic lateral sclerosis* (ALS). Numerous OL functions are disrupted in ALS, including OL differentiation, myelination trophic and support of axons, which could benefit from direct gene therapy action on OLs, provided that one finds a relevant transgene to be tested. In the SOD1^G93A^ mouse model of ALS, an AAV9-MBP-MCT1 vector injected icv at P10 failed to rescue the pathological phenotype [341]. MCT1 mRNA levels were increased in spinal cord and brain tissue, but counts of transduced OLs were not performed, leaving specific OL transductions unknown. The observed lack of effect was unexpected, particularly because the two-fold increase in MCT1 protein at disease onset was beneficial in another mutated SOD1 mouse ALS model [372].

#### 6.2.4. Several Non-Mendelian CNS Diseases

Several CNS diseases in the absence of a single gene defect may respond only to a gene addition strategy. This strategy requires the expression of genes that are not physiologically present in OLs but can improve neuronal health or inhibit disease mechanisms. The choice of such a therapeutic transgene is challenging and can be guided by previous experiments using neurotropic or astrotropic vectors to deliver growth factors or cytokines or enzymes by transduced OLs. Compared with neurons or astrocytes, OLs might produce higher amounts of growth factors or cytokines, which could be tested in mouse or NHP models. An already mentioned difficulty is the need to reach OLs in the spatially defined regions specifically affected by the diseases [373]. AD and neuropsychiatric diseases are associated with hypometabolism [374], white matter [375,376] and myelin abnormalities [377]. For this reason, improving the oligodendroglial oxidative metabolism of OLs could help prevent irreversible axon degeneration.

*Multiple System atrophy (MSA)*. Prominent mechanisms involve synuclein-associated protein aggregates that accumulate in OLs. These aggregates are thought to elicit changes in OL function, such as reduced neurotrophic support and demyelination, leading to neurodegeneration. To reverse this deleterious cascade, gene therapy should introduce a therapeutic gene able to support failing cell functions.

*Parkinson’s disease (PD).* We have previously seen that the neuron-targeted delivery of genes encoding GDNF, NRTN, BDNF and CDNF could yield positive results in mouse models. It would be worth testing whether such production could be increased using OLs instead of neuron transduction. Transduced OLs could also act through their myelin-independent role in supporting glutamate signaling [378], potentially impacting neurodegeneration [379]. Mounting evidence suggests that neuron-released protein aggregates are central to microglial activation, which in turn orchestrates neuroinflammatory processes potentially harmful to neurons [380]. The production of anti-inflammatory cytokines by transduced OLs could be used to inhibit pathomechanisms linked to innate inflammation [381]. Two ideal objectives of gene therapy in PD would be to reduce the accumulation of protein aggregates by using OLs to influence the metabolic and internal milieu of neurons or to improve dopamine synthesis using the transduction of OLs, but finding transgenes to achieve this goal seems currently out of reach.

*Alzheimer’s disease (AD)*. We previously discussed the deleterious role that OLs seemed to play in AD. The structural integrity of myelin deteriorates with age, and this deterioration appears to be exacerbated in AD. A recent study placed myelin defects upstream of amyloid-β (Aβ) plaque formation [382]. Targeting OLs to counteract mechanisms such as neuroinflammation [383] or neuronal dysmetabolism in the hippocampus are potential objectives. Given the pathogenic role of protein aggregates in AD [74], as in PD, it is difficult to find a gene whose expression in OLs would reduce the accumulation of these aggregates in affected hippocampal neurons. Increasing the supply of energy fuels to neurons and axons could be an objective of gene therapy since AD is associated with an energy deficit [49]. The reduction in OL MCT1 that occurs with aging may increase the risk of axonal degeneration in NDDs, including AD [384]. Transducing OLs with the *GDNF, BDNF* or *NGF* genes may provide a local source of these factors that are altered in AD [332,385]. Cholesterol regulation in OLs could be another target of gene therapy [123]. Since OLs contribute to Aβ plaque formation in AD through highly expressed amyloidogenic genes, identifying the mechanisms that slow down Aβ generation in OLs would pave the way for novel gene therapies [386]. The complement system is a tightly regulated innate immune system playing a key role in regulating CNS function and development. C5aR1, the receptor of the C5a complement factor, is expressed by clusters of OLs during the complex cellular phase of AD initiated by C5aR1-induced secreted mediators [387]. C5aR1 inhibition reduces plaque load, gliosis and memory deficits in animal models and could thus serve as a target for gene therapy of OLs [388].

*Multiple sclerosis* (MS) is another disease that could benefit from OL-targeting gene therapy.

Surprisingly, gene therapy has not been mentioned in recent reviews of innovative MS treatments or remyelination strategies [389,390,391,392,393], and new attempts have used AAV vectors in animal models [394], which are poorly representative of the human disease [395]. In active demyelinating lesions, the preserved number of mature OLs suggests a relative preservation of OLs, in contrast to the significant loss of OLs in the chronic lesions linked with disease progression [396,397,398]. This supports that mature OLs could be used as target cells for gene therapy in the initial stage of relapsing MS. A variety of transgenes could be tested at this stage to promote OL survival and limit immune-mediated CNS demyelination [399]. One could, again, try to improve energy metabolism of OLs [137]. Mature OLs have recently been shown to promote remyelination in MS [32,400,401,402,403].


*Psychiatric diseases*


It is not surprising to the reader of this review that OLs were not mentioned in a recent article dealing with gene therapy of rare psychiatric disorders [404]. Indeed, few people would postulate that therapeutic approaches of psychiatric diseases, such as autism, schizophrenia or depression, would one day include gene therapy. However, for researchers who think that the pathogenesis of psychiatric diseases involves defective molecular/cellular mechanisms, the idea of targeting brain cells, including OLs, with AAV vectors may not appear entirely unreasonable [145]. For example, since OLs may participate in the pathogenesis of depression [149] or schizophrenia [405], targeting these cells could one day become a novel strategy to treat severe and drug-resistant forms of these diseases.

#### 6.2.5. Genetic Axonopathies

*Spinocerebellar ataxia* (SCA) are associated with OL signatures [110]. Restoring myelination [366] or brain cholesterol turnover are two potential objectives of gene therapy [366,367].

*Hereditary spastic paraplegias* (HSPs)

Given the close interactions of OLs with the long and vulnerable axons of these tracts, transducing spinal cord OLs with metabolic transgenes could contribute to improving axonal health.

#### 6.2.6. *Hypoxic–Ischemic Diseases*

*Brain ischemia–reperfusion injury* affects OLs through oxidative stress, inflammation, failure of mitochondrial energy source and excitotoxicity, generating demyelination, axonal function and survival [154,155]. Some of these processes could be targeted by OL gene therapy.

*Vascular dementia* (VaD) involves a variety of neuronal and vascular lesions, leading to oxidative stress, inflammatory damage and demyelination, and it is closely associated with dysfunction of OLs [156].

*White matter damage* (WMD) in premature neonates. The fact that the BBB is permeable to AAV vectors in premature newborns creates an opportunity to use AAV vectors. A gene therapy, targeting mature OLs in brain regions prone to WMD might reduce or prevent white matter lesions [406]. A major challenge in developing proof-of-concept studies is that no animal model can actually mimic the human situation [407].

#### 6.2.7. Injury of Spinal Cord

Traumatic spinal cord injury triggers OL necrosis and apoptosis, with apoptosis continuing at the chronic stages of evolution. Loss of OLs causes demyelination and impairs axon function and survival [157,158]. Two OL populations, MOL2 and MOL5/6, could be more specific targets of gene therapy [37].

#### 6.2.8. Brain Regeneration Using Neuronal Precursor Cell Grafts

At the advanced stage of PD, AD and HD evolution, neuronal loss occurs in spatially and functionally defined brain regions. Since the mammalian brain cannot regenerate neurons, the only possible rescue in this situation is the grafting of neuronal progenitor cells (NPCs) to repopulate the affected region [408,409,410]. However, there is a major hurdle to this approach: the immune rejection of the allogeneic graft and the need for lifelong pharmacological immunosuppression, which has serious adverse effects. To avoid or alleviate immune rejection, research has focused on generating hypo-immune NPCs through genetic engineering [411,412]. Our group is developing a different strategy using AAV gene therapy to target the OLs located around and inside the zone of neuronal loss with a transgene encoding an anti-immuno-inflammatory cytokine to inhibit the allogeneic rejection of the grafted NPC.

## 7. Distinctive Challenges of OL-Targeting Gene Therapy

Just because targeting OLs is a new strategy, this does not mean that it will ensure the success of gene therapy in CNS diseases. The technical and biological challenges are numerous.

Technically, it will be necessary to develop highly oligotropic capsids and promoters that strongly express therapeutic transgenes in OLs. Promising examples of both have been cited in Section 6.1. The technology for creating recombinant capsids capable of prioritizing the targeting of cell populations is under active development [193]. To our knowledge, OLs have not yet been included among the tested cell types, but this is likely to occur when OLs become targets of choice. For the design of promoters activated in OLs, this will be achieved by manipulating the promoters or other regulatory sequences of myelin synthesis genes, an example of which is shown in Figure 1.

The biological challenges are more formidable and will rely on OL functions capable of interfering with the pathophysiological mechanisms of CNS diseases. In several areas, transducing OLs with AAV vectors may play an original and distinctive role. The first is the restoration of biochemical pathways specific to OLs. Another one is the synthesis of myelin and sheath enwrapment for CNS diseases that involve defects in myelination, such as genetic leukodystrophies, WMD of prematurity, and diseases like AD or HD that show defects in white matter in their early stages. A third field of action for transduced OLs is the supply of energy fuels to axons, since axons suffer from an energy deficit in many diseases such as AD [42] or other NDDs of aging [374]. As alterations in brain cholesterol homeostasis are linked to neurodegeneration, another metabolic objective could be to modify OL cholesterol metabolism and local cholesterol load white matter, for example in MS [413] or HD [370] or SCA [367]. Another approach is to make OL factories of growth factors, chemokines or cytokines. AAV gene therapy expressing growth factor genes, in neurons or astrocytes, ultimately had only a limited clinical effect, described in Section 6. OL transduction might provide stronger local trophic support to neurons through a more abundant production of growth factors [414]. OL production of anti-inflammatory cytokines may limit neuroinflammation in MS, AD or other diseases. OL production of anti-inflammatory cytokines, such as IL10, could also combat rejection of grafted NPCs by the local immune system [411]. Other major pathophysiological mechanisms will not be easy to counteract through OL targeting, such as the prion-like propagation of protein aggregates in synucleinopathies or the defect in dopamine synthesis in neurons of the nigro-striatal circuit. There are still many unknowns in the temporal sequence and interactions of the pathological mechanisms responsible for PD [115], AD [415] or other CNS diseases [416]. Shared alterations in gene expression in OLs across pathologies might indicate pathology-associated pathways that have not yet been explored in the search for therapies [69]. In the near future, new pathogenic mechanisms will likely be unraveled at the molecular and cell level, thanks to the connectome, interactome and metabolome approaches currently developed in PD [417,418,419,420,421,422,423,424], AD [121,425,426,427,428,429], HD [430] and MS [400]. These studies may reveal new avenues for OL gene therapy, given the multiple cross-talks between OLs and neurons, axons, astrocytes, microglia, ependymal and endothelial cells.

In conclusion, OLs deserve to be targets more often for gene therapy in a variety of CNS disorders, given their distinctive functions and properties (Table 3). This is the case for diseases caused or aggravated by OL dysfunctions. Therapeutic transgenes expressed in OLs can replace or silence defective OL genes or counteract pathological mechanisms occurring in other cells. During the early phase of brain or spinal cord diseases, the multiple crosstalks of transduced OLs could protect the CNS from defects in myelination, metabolic failure and neuroinflammation. In advanced stages of CNS diseases, when neuronal loss has occurred, surviving OLs might still express genes that prepare brain regions for NPC grafting.

## Figures and Tables

**Figure 1 cells-13-01973-f001:**
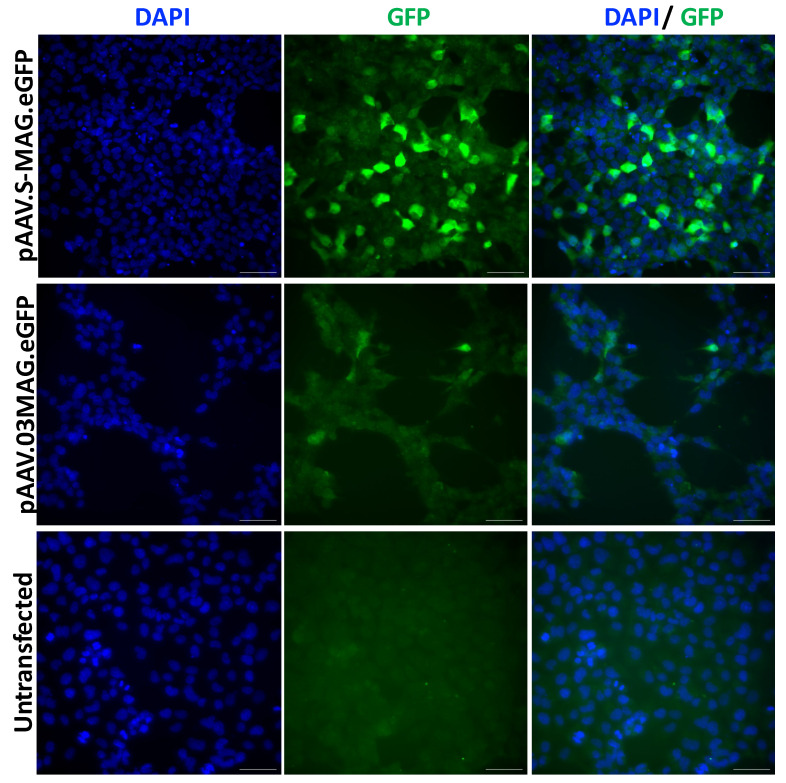
HOG cell line transiently transfected with eG FP expressing AAV plasmids under the control of Synthetic MAG promoter (pAAV. SMAG eG FP), or 0.3 kb MAG promoter. 48 h post-transfection, eG FP expression was analyzed by immunofluorescence. Scale bar: 50 cm. Fluorescent images were acquired using a Leica DM6 B fluorescence microscope (Leica, Wetzlar, Germany). Post-acquisition image analysis was performed using ImageJ software (latest v. 1.54, National Institutes of Health, Bethesda, MD, USA).

**Table 1 cells-13-01973-t001:** AAV gene therapy attempts at targeting neurons or astrocytes in animal models of CD.

Reference	Animal Model	Capsid	Promoter	Gene	Cell Tropism	ROA (vg/kg)	Age	Results (Age)
Matalon, 2003 [288]	AKO mouse	rAAV	CMV	hASPA	?	i-p ^a^	M3	ASPA expression (M6)
Klugmann 2005 [289]	CD rats tm rats	rAAV	CBA	hASPA	Neurons	i-p ^b^	P22	ASPA expression
Gao 2013 [16]	AKO mouse	AAV9	CBA	hASPA	Neurons	iv	P0-20	Motricity ± (M6)
Gao * 2017 [290]	AKO mouse	AAV9	GFAP	hASPA cod-opt	Astrocytes not OL	iv	P1	Motricity, fMRI, histology
Bannerman 2018 [291]	nur7 mouse	AAV2-8	U6	Nat8l shRNA	Neurons (<1% Astro, OL)	icv i-cist	P1	Motricity +
Klugmann 2022 [292]	AKO	Cy5	Dual U6 Mbp	Nat8l-shRNA hASPA	Neurons	i-p ^c^	M3	Motricity ++ Biochemistry ++
Beard-BB * 2019 [17]	Mac. Fasc	AAV9	CBA	hASPA cod-opt	?	iv 1.8 × 10^−14^	2.5 yrs	6 weeks later vgc whole CNS

i-p: intraparenchyma, ^a^ striatum, thalamus. ^b^ cingulum, internal capsule. ^c^ thalamus, striatum, cerebellum; i-cist: intracisternal, icv: intracerebroventricular. AKO: ASPA knockout. * vector later used in Ph1 trials.

**Table 2 cells-13-01973-t002:** Oligotropic AAV gene therapy in mouse models of CD.

Lab	Animal Model	AAV Capsid	Promoter	Gene	Cell Tropism	ROA	Age	Results (Age)
Francis * 2016 [188]	nur7 mouse	Olig001	CBh	hASPA	OL	i-p ^b^ icv	P2-3	CD prevention (M3)
Von Jonquieres 2018 [364]	AKO mouse	AAV2	Mbp-WPRE	hASPA	OL	i-p ^c^	P30	hASPA in OL (M6)

^b^ intracingulum and cerebellum ^c^ striatum, thalamus, cerebellum; * vector later used in Ph1 trials.

**Table 3 cells-13-01973-t003:** Functions and properties that make OLs interesting candidates for AAV-mediated gene therapy of CNS diseases.

Slow turnover that guarantees the durability of expression of therapeutic transgenes; Myelination of axons, myelin maintenance, remyelination of demyelinated lesions; Supply of energetic substrates to neurons/axons; Transfer of signaling molecules to axons; Multiple crosstalks with astrocytes and microglia; Improvement in OL trophicity and survival; Contribution to CNS immune and inflammatory homeostasis; Survival of OL in regions affected by neuronal loss; Capacity to release transgene products in affected regions of the brain; Induction of regional tolerance to neuronal stem cell grafts.

## Data Availability

Not applicable.

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
