# Peer review of "Oligodendrocytes, the Forgotten Target of Gene Therapy"

_cells, 2024, doi:10.3390/cells13231973_

Round 1

Reviewer 1 Report

Comments and Suggestions for Authors

The review article “Oligodendrocytes, the forgotten target of gene therapy” by Yasemin et al is needed by the field, as there are few if any reviews that take on this, aiming to make the case for giving OL their rightful place in CNS gene therapy.

Comment 1. The review could use some work to make it easier to digest, both in the text and visually, something like that interactions between OL and other cell types should be very informative.

Comment 2: In the abstract and text, the author described a simple explanation why have OL been left out of most gene therapy attempts as that the CAG, CBh, and CMV promoters widely used in gene therapy studies, and considered ubiquitous, seemed unable or poorly able to activate the transcription of episomal transgene copies within OL, for unknown biological reasons. I think there are many more other reasons than that, like the not clear roles of OL in many neurological diseases, much less neurological diseases with the initial pathology in OL, fewer researchers having the financial support, etc.  

Comment 3: There are many typos and editing works that need to be fixed/done. To finalize the paper, the author could ask a native English speaker to perform proof reading.

Comment 4: There are so many abbreviations in the review article with some are not commonly used. The abbreviation with the full name should occur the first time shown in the review. A table with all abbreviations could help readers to follow.

Comment 5: It would be nice to see some proposals on how to improve gene therapy in OL, including more funding, more research on OL, better collaboration, etc. How would the authors propose to increase interest in the OL study so that we could target OL better when designing gene therapy to treat diseases involving OL pathophysiology.  

Comments on the Quality of English Language

see comments to the authors

Author Response

The review article “Oligodendrocytes, the forgotten target of gene therapy” is needed by the field, as there are few if any reviews that take on this, aiming to make the case for giving OL their rightful place in CNS gene therapy. 

We thank the Reviewer for his positive comments. We do not know about any other review article tackling this subject. For this reason, we had to go through a large piece of literature in order to acknowledge and summarize the many studies i) supporting a major role of OL in CNS health and disease ii) having attempted to use gene therapy in CNS diseases. This explains why the review article is rather long. Even if we could not cite, understandably, all CNS diseases.

Comment 1. The review could use some work to make it easier to digest, both in the text and visually, something like that interactions between OL and other cell types should be very informative.

We agree with the Reviewer that some work was needed to be easier to digest. We provided a Graphical Abstract and worked on the ms to limit, if not completely avoid, redundancies. We made a major effort to re-write and clarify many paragraphs, and to make the text more fluid.

Comment 2: In the abstract and text, the author described a simple explanation why have OL been left out of most gene therapy attempts as that the CAG, CBh, and CMV promoters widely used in gene therapy studies, and considered ubiquitous, seemed unable or poorly able to activate the transcription of episomal transgene copies within OL, for unknown biological reasons. I think there are many more other reasons than that, like the not clear roles of OL in many neurological diseases, much less neurological diseases with the initial pathology in OL, fewer researchers having the financial support, etc.

We fully agreed with this comment and we corrected the new version accordingly in the Abstract, Introduction and Discussion. We have not discussed, though, the financial support of gene therapy research, which is currently squeezed by reduced investment in the sector (hopefully transiently) (see High, K.A. Gene and cell therapy in 2023: Rich pipeline, slimming resources? Mol Ther 2024, 32, 3-4, doi:10.1016/j.ymthe.2023.12.003). Indeed, we felt this would have been beyond the scope of the review article and would lead us to general comments regarding gene therapy

Comment 3: There are many typos and editing works that need to be fixed/done. To finalize the paper, the author could ask a native English speaker to perform proof reading

We have carefully fixed typos and English mistakes, and asked a native English speaking colleague to correct our writing.

Comment 4: There are so many abbreviations in the review article with some are not commonly used. The abbreviation with the full name should occur the first time shown in the review. A table with all abbreviations could help readers to follow.

We checked the full name when the abbreviation occur the first time in the ms. A table would take a large place, but we can provide one if needed.

Comment 5: It would be nice to see some proposals on how to improve gene therapy in OL, including more funding, more research on OL, better collaboration, etc. How would the authors propose to increase interest in the OL study so that we could target OL better when designing gene therapy to treat diseases involving OL pathophysiology.

Our view is that the biology of OL is being investigated very thoroughly, for example by groups such as that led by Klaus Armine Nave, and has been producing many seminal observations through highly cited studies. And also for disease mechanisms. More research on OL is obviously desirable, but this is also true for many fields of molecular and cell biology. Our point herein is that the articulation between basic OL biology research, disease mechanisms and gene therapy attempts needs to be improved. Testing therapeutic vectors inspired by basic research should undoubtedly be more active, because of the enormous potential of OL as key players in CNS health and diseases. We do not believe that « more funding » should be discussed here. In summary, we do hope that this review article will contribute to give researchers a boost for testing new therapies based on OL transduction.

Reviewer 2 Report

Comments and Suggestions for Authors

This is well composed and articulated review about the function of oligodendrocytes and their role in neuropathology and pathogenesis of CNS diseases. The review also provides a comprehensive discussion of AAV-based therapies targeting oligodendrocytes. 

Missing from the manuscript is a list of challenges facing oligodendrocyte-based therapies and the potential use of OL progenitors as a therapeutic approach. This review could also benefit from some references and a brief discussion of oligodendrocyte in the optic nerve and in the pathogenesis of optic neuropathies as well. 

Author Response

This is well composed and articulated review about the function of oligodendrocytes and their role in neuropathology and pathogenesis of CNS diseases. The review also provides a comprehensive discussion of AAV-based therapies targeting oligodendrocytes. 

We thank the Reviewer for his positive comments. We have further enriched our article by reporting and commenting additional studies.

Missing from the manuscript is a list of challenges facing oligodendrocyte-based therapies and the potential use of OL progenitors as a therapeutic approach. This review could also benefit from some references and a brief discussion of oligodendrocyte in the optic nerve and in the pathogenesis of optic neuropathies as well. 

We fully agree with the Reviewer. We have written a full paragraph to list and comment the challenges facing OL-based gene therapy. We have followed his suggestion of discussing optic neuropathies.

Reviewer 3 Report

Comments and Suggestions for Authors

The Authors presented a Review with the title "Oligodendrocytes, the forgotten target of gene therapy", enlighting for sure the interest and curiosity of readers. In fact, the issue discussed in the Review is a hot point in the gene therapy panorama and the Review could be a milestone for Researchers in the field.

Unfortunately, the overall presentation of concepts in the Review was fragmented, with a basic report of information found in scientific papers, but not connected by a logical path. Most importantly, the major part of the Review is a descripting reporting of data regarding CNS-targeting AAV gene therapy approaches not involving oligodendrocytes, from which does not emerge the reason why this cell type is excluded from scientific reasoning. On the other hand, the Review is dotted with personal considerations of the Authors, that are not supported by any further explanation. Finally, the scientific language used is poor, with repetitive use of the same words, often associated with a lack of details.  

The text must be deeply revised, please some of the major points in the list below.

Line 11: Could express instead of be able to express.

Line 36: Reference is missing.

Line 45-54: This text is the same as the abstract. Please rephrase one of the two, possibly the Introduction, making it more informative and more detailed than the abstract-reported counterpart.

Line 61: and TO the NHP marmoset.

Line 62-65: please rephrase making a single, clearer sentence.

Line 68-69: again, the same sentence as in the abstract. please rephrase.

Line 74-75: please pay attention to numbers and better explain their origin. The 17.4 millions of the first sentence in which ratio is with the 21-29 billions of the second sentence? 20% is referring to what tissue of what species?

Line 89: detects.

Line 93-96: please rephrase this sentence for the sake of clearance and readability.

Line 98-105: please rephrase making the sentence more connected and fluid to easily express the logical path of all the elements presented.

Line 110: maybe the Authors means the understanding, rather than the treatment.

Line 126: I suggest to introduce subparagraphs to better identify the diseases of interest.

Line 127: please rephrase: heterogeneous? in terms of?

Line 128-129: please close the second pair of brackets and add in the brackets that these diseases are examples.

Line 129-131: please rephrase.

Line 135: better “for which the molecular/cell substratum, if present, is still unknown”.

Line 137: please rephrase in a sentence, not a question.

Line 147: “vis-à-vis axons”, it is not clear what these words means in the context of the sentence.

Line 147-148: this sentence is not fully understandable, probably it lacks details and description.

Line 151: please choose: “is implicated in a wide number” or “contributes to a wide number”.

Line 154 Strictu sensu in italic.

Line 158: “synthesized, in”, remove comma.

Line 162: “to contribute or TO aggravate”.

Line 163: correct “even if it is not recognized as”.

Line 164: please specify to what this "they" is referring.

Line 170: correct "or the result of" or " or it results from"; add “alpha” where necessary.

Line 176: brain homogenates?

Line 183-186: please add reference.

Line 188: TO glia.

Line 190: what do the Authors mean for "inflammation reprogramming"?

Line 191: please add at least a sentence with some inherent information regarding OL and AD.

Line 212: I suggest to introduce subparagraphs to better identify the considered elements.

Line 217: please correct “in particular” with “briefly”.

Line 219: “contingent”, do the Authors mean "dependent on"?

Line 234: “alternately” correct alternatively.

Line 240-241: add reference.

Line 244-245: please rephrase in a clearer sentence, for example: However, the precise translation of the best treatment timing from animal models to patients it is often inapplicable.

Line 248: What do the Authors mean with "general population"?         

Line 248-249: Please rephrase, word duplication makes it difficult to understand the meaning of this sentence.

Line 251: Capsid reach the cells, the serotype can influence this ability. Please make it understandable.

Line 272: “to test its suitability”, It is not a direct assessment of suitability for the human target, please be more precise: to better predict its potential translatability to humans.

Line 273-274: This sentence is not connected with the previous one.

Line 285-286: Add references.

Line 328: Short sentences all starting with the same word, please connect them in a single sentence.

Lie 330-332: Rephrase this sentence, info have already been presented in the previous one.

Line 333-337: See the comment above on short sentences.

Line 355: remove bracket.

Line 355-356: Part of the sentence is missing/wrong. Please change “where” with "of".

Line 359-360: Is this the title or the first sentence? Reference is lacking.

Line 377: “in juvenile..macaques.” Please correct. This sentence can not be self-standing.

Line 395: “(maybe studied at a too young age)” Is this an observation of the Authors? Please better contextualize.

Line 401: remove bracket.

Line 420: “experimental creativity” I would rather say something like "the heterogeneity of experimental applications in this field".

Line 421: please correct “disease” with “patients”.

Line 424-425: add reference.

Line 426: “earlier” repetition, correct.

Line 455-464: These information can be presented as a Table with the respective references or at least a precise indication of the tools used to produce the presented numbers.

Line 470: “not supported by any rationale”, I suppose this is an opinion of the Authors, if it is so, please better explain the facts behind this sentence, or remove it.

Line 482: “thus supposedly”, Maybe the Authors mean "thus supposedly also in in human"?

Line 486: Post vector what? Transfection?

Line 499: “nur”, correct.

Line 588, Fig.1: The scale is too little to be seen and the images appear out of focus. Please add information regarding the acquisition of these images or the respective reference.

Line 620: That is an obvious point that is moving research in the field since decades. Please add more information or remove this sentence.

Line 694-695: Still, there is a list of examples where OL have been excluded from research experiments, but no in-depth analysis of the reasons why this happened is presented. And again, here is an obvious consideration that can not be standalone in the absence of the above information.

Comments on the Quality of English Language

The use of English language is poor, with short sencences that often appear not connected, but rather reported in a fragmented way, impacting on the readability and quality of the Review. 

Author Response

The Authors presented a Review with the title "Oligodendrocytes, the forgotten target of gene therapy", enlighting for sure the interest and curiosity of readers. In fact, the issue discussed in the Review is a hot point in the gene therapy panorama and the Review could be a milestone for Researchers in the field.

We thank the Reviewer for these positive comments.

Unfortunately, the overall presentation of concepts in the Review was fragmented, with a basic report of information found in scientific papers, but not connected by a logical path.

This opinion contrasts with that of Reviewer 2 « This is a well composed and articulated review about the function of oligodendrocytes and their role in neuropathology and pathogenesis of CNS diseases. The review also provides a comprehensive discussion of AAV-based therapies targeting oligodendrocytes ». Nevertheless, we understand and respect Reviewer 3’s opinion, and agree the presentation of concepts sould be improved in the new version. We have worked intensively to achieve this goal.

Most importantly, the major part of the Review is a descripting reporting of data regarding CNS-targeting AAV gene therapy approaches not involving oligodendrocytes, from which does not emerge the reason why this cell type is excluded from scientific reasoning.

We tried to have this reason emerging more strongly from the revised ms. We thought we needed to cite and summarize neurotropic, astrotropic, and microglia-tropic approaches, which have been used in the vast majority of gene therapy attempts until now. Doing so carries the risk of a description that is only reviewing data from previous gene therapy studies. This is why we have taken Reviewer 3’s opinion very seriously and have largely re-worked our ms, trying to ameliorate the comprehensive discussion of OL-based gene therapy. It is a fearful challenge to somewhat underscore the limits of transducing neurons, astrocytes, microglia that represent >90% of existing data of CNS gene therapy research and put them in the perspective on OL gene therapy studies that don’t yet exist. The article should not give the impression of naively surpassing the work of highly recognized experts. That’s why the spirit of this review is simply to try to open a new door and explore what’s behind, which is largely unknown. The reason why OL have been excluded from gene therapy attempts (not scientific reasoning) should actually be asked to previous experts. It has no rationale, in a way, other than those presented in the Abstract and discussed in our ms (and commented by Reviewer 1) : lack of curiosity, limits or lack of established knowledge about OL involvement in many major CNS diseases, OL being wrongly considered resistant to AAV transduction (in fact a promoter problem), studies out of the box considered scientifically and financially risky vs more classic other cell types. We did our best, however, to follow the proposition of the Reviewer.

On the other hand, the Review is dotted with personal considerations of the Authors, that are not supported by any further explanation.

We agree, but we thought that personal considerations could stimulate OL-based gene therapy attempts. In this respect, and in the almost total absence of « reviewable » OL-based studies, we thought we should express personal views in this review article. Actually, this was our only option when we engaged in speaking about potential yet non-existing experiments. It is a bit harsh to say that our propositions were not supported by any explanation. May we respectfully cite Reviewer 2 who found our discussion « comprehensive ». In the new version of our article, we have nevertheless worked intensely to provide a scientific rationale to our proposition of testing OL potential in gene therapy and try to respond to Reviewer 3’s critics

Finally, the scientific language used is poor, with repetitive use of the same words, often associated with a lack of details.  

We worked hard on the revised version to avoid these errors.

The text must be deeply revised, please some of the major points in the list below.

Line 11: Could express instead of be able to express.

Line 36: Reference is missing.

Line 45-54: This text is the same as the abstract. Please rephrase one of the two, possibly the Introduction, making it more informative and more detailed than the abstract-reported counterpart.

Line 61: and TO the NHP marmoset.

Line 62-65: please rephrase making a single, clearer sentence.

Line 68-69: again, the same sentence as in the abstract. please rephrase.

Line 74-75: please pay attention to numbers and better explain their origin. The 17.4 millions of the first sentence in which ratio is with the 21-29 billions of the second sentence? 20% is referring to what tissue of what species?

Line 89: detects.

Line 93-96: please rephrase this sentence for the sake of clearance and readability.

Line 98-105: please rephrase making the sentence more connected and fluid to easily express the logical path of all the elements presented.

Line 110: maybe the Authors means the understanding, rather than the treatment.

Line 126: I suggest to introduce subparagraphs to better identify the diseases of interest.

Line 127: please rephrase: heterogeneous? in terms of?

Line 128-129: please close the second pair of brackets and add in the brackets that these diseases are examples.

Line 129-131: please rephrase.

Line 135: better “for which the molecular/cell substratum, if present, is still unknown”.

Line 137: please rephrase in a sentence, not a question.

Line 147: “vis-à-vis axons”, it is not clear what these words means in the context of the sentence.

Line 147-148: this sentence is not fully understandable, probably it lacks details and description.

Line 151: please choose: “is implicated in a wide number” or “contributes to a wide number”.

Line 154 Strictu sensu in italic.

Line 158: “synthesized, in”, remove comma.

Line 162: “to contribute or TO aggravate”.

Line 163: correct “even if it is not recognized as”.

Line 164: please specify to what this "they" is referring.

Line 170: correct "or the result of" or " or it results from"; add “alpha” where necessary.

Line 176: brain homogenates?

Line 183-186: please add reference.

Line 188: TO glia.

Line 190: what do the Authors mean for "inflammation reprogramming"?

Line 191: please add at least a sentence with some inherent information regarding OL and AD.

Line 212: I suggest to introduce subparagraphs to better identify the considered elements.

Line 217: please correct “in particular” with “briefly”.

Line 219: “contingent”, do the Authors mean "dependent on"?

Line 234: “alternately” correct alternatively.

Line 240-241: add reference.

Line 244-245: please rephrase in a clearer sentence, for example: However, the precise translation of the best treatment timing from animal models to patients it is often inapplicable.

Line 248: What do the Authors mean with "general population"?         

Line 248-249: Please rephrase, word duplication makes it difficult to understand the meaning of this sentence.

Line 251: Capsid reach the cells, the serotype can influence this ability. Please make it understandable.

Line 272: “to test its suitability”, It is not a direct assessment of suitability for the human target, please be more precise: to better predict its potential translatability to humans.

Line 273-274: This sentence is not connected with the previous one.

Line 285-286: Add references.

Line 328: Short sentences all starting with the same word, please connect them in a single sentence.

Lie 330-332: Rephrase this sentence, info have already been presented in the previous one.

Line 333-337: See the comment above on short sentences.

Line 355: remove bracket.

Line 355-356: Part of the sentence is missing/wrong. Please change “where” with "of".

Line 359-360: Is this the title or the first sentence? Reference is lacking.

Line 377: “in juvenile..macaques.” Please correct. This sentence can not be self-standing.

Line 395: “(maybe studied at a too young age)” Is this an observation of the Authors? Please better contextualize.

Line 401: remove bracket.

Line 420: “experimental creativity” I would rather say something like "the heterogeneity of experimental applications in this field".

Line 421: please correct “disease” with “patients”.

Line 424-425: add reference.

Line 426: “earlier” repetition, correct.

Line 455-464: These information can be presented as a Table with the respective references or at least a precise indication of the tools used to produce the presented numbers.

Line 470: “not supported by any rationale”, I suppose this is an opinion of the Authors, if it is so, please better explain the facts behind this sentence, or remove it.

Line 482: “thus supposedly”, Maybe the Authors mean "thus supposedly also in in human"?

Line 486: Post vector what? Transfection?

Line 499: “nur”, correct.

Line 588, Fig.1: The scale is too little to be seen and the images appear out of focus. Please add information regarding the acquisition of these images or the respective reference.

Line 620: That is an obvious point that is moving research in the field since decades. Please add more information or remove this sentence.

Line 694-695: Still, there is a list of examples where OL have been excluded from research experiments, but no in-depth analysis of the reasons why this happened is presented. And again, here is an obvious consideration that can not be standalone in the absence of the above information.

We deeply thank Reviewer 3 for this meticulous analysis of our ms. We have almost never had the chance of such a precise review of our articles, and this helped us a lot for improving our writing. We have followed all underlined points, made the cited corrections, incorporated Reviewer’s own words. Most of all, we felt that the revised version is largely improved and we hope that Reviewer 3 would agree.

Round 2

Reviewer 3 Report

Comments and Suggestions for Authors

The Authors worked in the right direction to improve the quality of the paper.